



# Smoothing data series by means of cubic splines: quality of approximation and introduction of an iterative spline approach

Sabine Wüst [1], Verena Wendt [1,2], Ricarda Linz [1], Michael Bittner [1,3]

5 [1]Deutsches Fernerkundungsdatenzentrum, Deutsches Zentrum für Luft- und Raumfahrt, 82234 Oberpfaffenhofen, Germany

[2]Umweltforschungsstation Schneefernerhaus, Zugspitze, Germany, now at: Institut für industrielle Informationstechnik, Hochschule Ostwestfalen-Lippe, Ostwestfalen-Lippe, Germany

[3]Institut für Physik, Universität Augsburg, 86159 Augsburg, Germany

*Correspondence to*: Sabine Wüst (sabine.wuest@dlr.de)

10 **Abstract.** Cubic splines with equidistant spline sampling points are a common method in atmospheric science for the approximation of undisturbed background conditions by means of filtering superimposed fluctuations from a data series. Often, not only the background conditions are of scientific interest but also the residuals—the subtraction of the spline from the original time series.

Based on test data sets, we show that the quality of approximation is not increasing continuously with increasing number of 15 spline sampling points / decreasing distance between two spline sampling points. Splines can generate considerable artificial oscillations in the data.

We introduce an iterative spline approach which is able to significantly reduce this phenomenon. We apply it not only to the test data but also to TIMED-SABER temperature data and choose the distance between two spline sampling points in a way that we are sensitive for a large spectrum of gravity waves.



## 1 Introduction

It is essential for the analysis of atmospheric wave signatures like gravity waves that these fluctuations are properly separated from the background. Therefore, particular attention must be attributed to this step during data analysis. Splines are a common method in atmospheric science for the approximation of atmospheric background conditions. The choice of a

sufficiently low number of spline sampling points ensures a smoothing of the original time series. Conclusions about the shortest wavelength / period which is still uniquely resolvable by the spline can be drawn from the application of Shannon's sampling theorem (see e.g. Gubbins, 2004) based on the number of spline sampling points used. Depending on the field of interest, either the smoothed data series or the residuals—the subtraction of a spline from the original time series—are further analysed (see for example the work of Kramer et al., 2016; Baumgarten et al., 2015; Zhang et al., 2012; Wüst and

Bittner, 2011; Wüst and Bittner, 2008; Young et al., 1997; Eckermann et al., 1995).

Algorithms for the calculation of splines are implemented in many programming languages and in various code packages making them easy to use. Nevertheless, spline approximations need sometimes to be handled with care when it comes to physical interpretation.

Figure 1 explains our motivation for the work presented below. It shows the squared temperature residuals averaged over

one year for the years 2010–2014 versus height between 44° N and 48° N and 5° O and 15° O (approximately 500 profiles per year). The vertical temperature profiles are derived from the SABER (Sounding of the Atmosphere using Broadband Emission Radiometry) instrument on board the satellite TIMED (Thermosphere Ionosphere Mesosphere Energetics Dynamics), data version 2.0 (details about this data version can be found in Wüst et al. (2016) and references therein, for example). For the calculation of the residuals, we applied a cubic spline routine with equidistant sampling points. The

distance of 10 km between two spline sampling points makes sure that we are sensitive for a large spectrum of gravity waves. Therefore, we take the squared temperature residuals as a simple proxy for gravity wave activity with vertical wavelengths up to 20 km. Obviously, the mean squared residuals do not only reveal a strong and continuous increase with height (note the logarithmic x-axis) as it is expected since gravity wave amplitudes should increase due to the exponentially decreasing atmospheric background pressure with altitude. Superimposed on this general increase of gravity wave activity

are well-pronounced oscillations with a wavelength of ca. 10 km, which is nearly equal to the distance between two spline sampling points.

Since we are not aware of any physical reason for this oscillation, we formulate the hypothesis that this is an artefact of the analysis. In order to avoid or at least reduce such problems we here propose an iterative variation of the cubic spline approach, which we explain in section 2. In section 3, we apply the original and the iterative approach to test data sets. The

results are discussed in section 4. A brief summary is given in section 5.



## 2 Methods and algorithms

The approach we investigate here relies on cubic splines with equidistant sampling points. Since spline theory is well-elaborated, we will therefore not go into much detail here. The algorithm we use is based on Lawson and Hanson (1974).

The first step for the adaption of a spline function to a data series on an interval [a, b] is the choice of the number of spline

sampling points (also called knots). These points divide the interval for which the spline is calculated into sub-intervals of equal length. For each sub-interval a third-order polynomial needs to be defined, that means the coefficients have to be determined. At the spline sampling points, not only the function value, but also the first and second derivatives of the two adjacent polynomials need to be equal. The optimal set of coefficients is calculated according to a least square approach where the sum of the squared differences between the data series and the spline is minimized.

According to the sampling theorem, the shortest wavelength / period which can be resolved by the spline is two times the distance between two consecutive spline sampling points. At the same time, this wavelength / period is the largest resolvable one in the residuals. Depending on the length of the data series only specific distances between two consecutive spline sampling points can be realised if the whole data series is approximated. The number of spline sampling points (and the length of the data series) therefore determines the sensitivity of the spline to specific wavelengths.

With SABER data we analysed temperature profiles which extend over a great height distance. We would like to operate the spline algorithm in the way that we provide independently of the ratio between length of the data series and number of spline sampling points the shortest wavelength which shall be resolved by the spline. That means that we have to cut the upper part of the profile from case to case. In detail, our spline algorithm works as follows. The scheme includes the iterative as well as

the non-iterative algorithm.

- *Step 1: Provision of shortest wavelength*

  We provide the algorithm the shortest wavelength which shall be resolved by the spline (in the following denoted as lim). It is equal to the doubled distance between two spline sampling points; therefore the distance between two

spline sampling points is equal to lim/2.

- *Step 2: Determination of x-values of the spline sampling points*

  The minimal x-value of the data series is subtracted from the maximal x-value, the difference is divided by lim/2. If the result is a whole-number, 1 is added. If this is not the case, the closest integer less than the result is calculated

and 1 is added. This is the number of spline sampling points used for the next step. It is denoted as n:

$$n = \left( \frac{x_{max} - x_{min}}{lim/2} - \frac{x_{max} - x_{min}}{lim/2} \bmod 1 \right) + 1 \tag{1}$$

  Knowing lim/2 and the minimal x-value, the x-values of the further spline sampling points can be calculated.





- *Step 3: Calculation of spline approximation*

  The spline approximation is calculated based on Lawson and Hanson (1974). If the length of the data series is not equal to an integer multiple of lim/2, the surplus part at the end of the data series is not subject of this step. For the non-iterative approach, the spline algorithm stops here.

- *Step 4 (only in the case of the iterative approach): Iteration of starting point*

  The first point of the data series is removed and step 2 and 3 are repeated. If the starting point is equal to the original minimal x-value plus lim/2, the algorithm proceeds with step 5.

10 - *Step 5 (only in the case of the iterative approach): Calculation of the final spline*

  The mean of all splines derived before is calculated. That is the final (iterative) spline.

For the iterative approach, the length of the data series is not the same in each iteration since data at the beginning and the end of the data series are not necessarily part of each iteration: at the beginning of the data series, this holds for all x-values

15 between the minimal x-value and the minimal x-value plus lim/2 (see step 4), at the end of the data series, this is the case for all values between the maximal x-value and the maximal x-value minus lim/2 (see step 3).

For the non-iterative approach, data are cut only at the end of the data series if the length of the data series is not equal to an integer multiple of lim/2.



## 3 Case studies

We generate a basic example using an artificial sine with a vertical wavelength of 3 km, a phase of zero and an amplitude of one. The function is sampled every 375 m (that means at its zero-crossings, at its extrema and once in between the zero-crossing and the next extremum / the extremum and the next zero-crossing).

The values for the sampling rate and the vertical wavelength are set arbitrarily. However, the spatial resolution of 375 m is motivated through the spatial resolution of TIMED-SABER, an instrument which is commonly used for the investigation of gravity waves (e.g. Zhang et al., 2012; Wright et al., 2011; Krebsbach and Preusse, 2007) and which delivered also the temperature profiles we used in fig. 1.

Fig. 2 (a) shows the test data series (dotted line) between 15 km and 100 km height. This great height range is chosen since it

facilitates the demonstration of our results. A (non-iterative) spline with a distance of 1.5 km between two spline sampling points is fitted (solid line). According to the sampling theorem, the chosen distance between two spline sampling points is small enough for resolving the oscillation in our test data. In part (b) and (c) of fig. 2, a spline with a distance of 1.6 km and 1.4 km between two spline sampling points is calculated. Part (d) to (f) of fig. 2 focus on the height range of 15 km to 50 km of fig. 2 (a) to (c): here, the height-coordinates of the spline sampling points are plotted additionally (dashed-dotted lines).

The asterisks mark the sampling points of the original sine. The spline adaption in fig. 2 (a) / (d) differs significantly from the spline adaption in fig. 2 (b) / (e) and 2 (c) / (f): apart from a slight oscillation at the beginning / end of the height interval, the spline is equal to zero in fig. 2 (a) / (d). The spline approximation plotted in fig. 2 (b) and (c) shows a beat-like structure in the whole height range.

In order to give an overview concerning the quality of adaption not only for some chosen examples as they were shown in

fig. 2, the test data set is approximated by a cubic spline with varying numbers of spline sampling points. The squared differences between the spline and the test data are summed up between 20 km and 40 km (this height interval is chosen in order to be consistent with fig. 7 later). We call this value the approximation error. It is not decreasing continuously with increasing number of spline sampling points / decreasing distance between two spline sampling points but it is characterized through a superimposed oscillation which reaches its maximum for a distance of ca. 1.5 km between two spline sampling

points (fig. 3, solid line). When changing the phase of the test data set to $\pi/2$ (instead of zero), the approximation error for a distance of ca. 1.5 km between two spline sampling points is much lower (fig. 3, dashed line), that means the sinusoidal oscillation is better adapted in this case. This makes clear that the approximation error depends on the phase of the oscillation (one can also say on the exact position of the spline sampling points).

The analysis described above is repeated, but the phase of the oscillation is varied between 0 and $2\pi$. The approximation

error (between 20 km and 40 km) is calculated for three different distances between two spline sampling points: 1.5 km (fig. 4, solid line), 1.4 km (fig. 4, short dashes) and 1.6 km (fig. 4, long dashes). The dependence on the phase is most pronounced for a distance of 1.5 km: the oscillation is adapted very well between 20 km and 40 km for a phase of $\pi/2$ and $3\pi/2$. For a phase of 0 and $\pi$, the contrary holds.





This example directly motivates the application of the iterative spline approach on the same test data set (see fig 5 (a)–(f) which can be directly compared to fig 2 (a)–(f): the black line represents the final spline approximation and the different colours refer to the spline approximations during the different iteration steps). In this case, the approximation error depends much less on the distance between two spline sampling points (fig. 6 (a)) and on the phase of the test data set (fig. 6 (b)).

5 Only for a distance of 1.6 km between two spline sampling points, a slight phase dependence is still visible (fig. 6 (b)).

Until now, we showed only test data which are not superimposed on a larger-scale variation like the atmospheric temperature background. Now, three sinusoidals with vertical wavelengths of 3, 5, and 13 km, phase 0, $\pi/3$, and $\pi/5$, and amplitude 0.5 (growing amplitude with height neglected for simplicity reasons) are superimposed on a realistic vertical temperature background (fig. 7 (a)). The background is based on CIRA-86 (COSPAR International Reference Atmosphere, Committee

10 on space Research and NASA National Space Science Data Center, 2006) temperature data for 45° N for January, which is brought on a regular grid using a cubic spline with a distance of 3 km between two spline sampling points. It was checked that no additional signatures are caused thereby. The approximation error shows three steps but no superimposed oscillations (fig. 7 (b)): the first step at ca. 6 km to 7 km (distance between two spline sampling points), the second one at ca. 2 km to 3 km and the last one at 1 km to 2 km. Following Shannon's sampling theorem, this observation can be explained through

15 the ability of the spline to adapt the original wavelengths. The realistic background makes clear why we restrict the calculation of the approximation error to the height range between 20 km and 40 km: this height interval is chosen in order to exclude especially the stratopause since the fast changing temperature gradient can cause additional problems for the spline approximation. Furthermore, the choice of this interval makes sure that the data used for fig. 7 (b) are part of each iteration step (which is not the case for the data at the beginning and the end of the data series, see section 2).





# 4 Discussion

In section 3, we showed that the ability of a spline to approximate oscillations varies

   a)     with the number of spline sampling points, and

   b)     with the exact position (height coordinate) of the spline sampling points.

While the first statement can be explained Shannon's sampling theorem, the second one might be surprising.

When the distance between two spline sampling points matches exactly half the wavelength of the test data, the approximation is worst for a phase of 0 and $\pi$. In this case, the spline sampling points are located exactly between the extrema of the test data. If the height coordinates of the spline sampling points agree with the height coordinates of the extrema of the test data, the contrary holds (in appendix A, we provide a mathematical explanation for this observation). The

dependence of the quality of approximation on the phase of the test data decreases with greater / smaller distances between two spline sampling points (fig. 3). These findings directly motivate the use of the presented iterative spline approach which is characterized by varying positions (height coordinate) of the spline sampling points.

Furthermore, we showed that if the distance between two spline sampling points is only slightly larger or smaller than half

the wavelength present in the data series and if enough wave trains are present (which might not be the case in reality), the (non-iterative) spline reminds of a beat (see fig. 2 (b) and (c), an explanation is given in appendix B). The subtraction of such a "beat" will lead to an artificial oscillation in the residuals with a periodically increasing and decreasing amplitude reaching ca. 70–80% of the original amplitude at maximum (fig. 2 (e) and (f)). This oscillation must not be interpreted as a gravity wave of varying amplitude, for example, and the described effect has to be taken into account when analysing wavelengths

similar to the doubled distance between two spline sampling points.

For our case studies, we used a constant and a realistic CIRA-based temperature background profile. We showed the approximation error (the squared differences between the spline and the test data summed up between 20 km and 40 km) decreases much smoother with increasing number of spline sampling points for the iterative approach compared to the non-

iterative one (compare fig. 3 to fig. 6 (a)) and the amplitude of the "beat"-like structure is reduced.

However, the motivation for this work was—as already mentioned—the results shown in fig. 1 which are characterized by a strong superimposed oscillation with a wavelength of approximately 10 km for which we do not have a physical explanation. Figure 8 (a) now depicts the mean squared residuals after the application of the iterative spline to the same data set, fig. 8 (b)

focuses on the year 2014 (the dashed line is based on the application of the iterative spline, the solid line refers to the non-iterative spline). This year is chosen arbitrarily and allows the direct comparison between the iterative and non-iterative approach. The amplitude of the superimposed oscillation is reduced significantly but the oscillation can still be observed. This supports our hypothesis that the strong superimposed oscillation described in fig. 1 is an artefact of the non-iterative





spline de-trending procedure. Furthermore, it becomes obvious now that gravity wave activity is less variable between approximately 45 km and 60 km height compared to the height range above. This is in accordance with literature (e.g. Mzé et al., 2014; Offermann et al., 2009). For most heights, the mean squared residuals are smaller for the iterative approach than for the non-iterative one. At 38 km height for example, the difference reaches ca. 2.5 K², which is approximately 32 %

(referring to the mean value of both approaches).

In order to give a comprehensive comparison of the iterative and the non-iterative spline algorithm, we calculate also the mean (non-squared) residuals. In this case, the results look very similarly; in both cases, they again show an oscillation with a vertical wavelength of 10−20 km (fig. 8 (c) for the non-iterative approach, fig. 8 (d) for the iterative spline approach). We

can explain this in the following way: when calculating the mean (non-squared) residuals and the mean squared residuals at a specific height, one refers to two different parameters of the distribution of residuals at that specific height. While the mean (non-squared) residuals estimate the mean of the distribution, the mean squared residuals refer to the variance of the distribution. We conclude: at a defined height, the iterative approach changes the mean of the distribution of the residuals only slightly, but it reduces its spread significantly. For individual profiles, the approximation through the iterative approach

is therefore less variable in average and can be recommended. The iterative approach can also be recommended if squared residuals are needed for further analysis (e.g. for the calculation of the wave potential energy). If non-squared residuals will be analysed, it does not make a difference in average which approach is applied. In this case, only waves with amplitudes larger than 0.5 K in the stratosphere and 1.0 K in the mesosphere (fig. 8 (c) and (d)) should be taken seriously.

It is known that height regions where the temperature gradient changes are challenging for approximation methods, and we speculate that this is at least one reason the superimposed oscillation in fig. 8 (c) and (d). However, analysing this hypothesis is beyond the scope of this manuscript.

There exist many methods to approximate / de-trend / filter time series (see e.g Baumgarten et al. (2015), and references

therein) and we do not claim that the presented iterative cubic spline is the best method for every purpose and every data series. It is just one possible algorithm which reduces disadvantages of the non-iterative cubic spline routine as it was proposed by Lawson and Hanson (1974) like the dependence of approximation on the exact position (height coordinate) of the spline sampling points. However, it comes along with enhanced computational effort which is of special importance when analysing large data sets.






## 5 Summary

It is essential for the analysis of atmospheric wave signatures like gravity waves that these fluctuations are properly separated from the background. Therefore, particular attention must be attributed to this step.

Cubic splines with equidistant sampling points are a common method in atmospheric science for the approximation of superimposed, large-scale structures in data series. The subtraction of the spline from the original time series allows the investigation of the residuals by means of different spectral analysis techniques. However, splines can generate artificial oscillations in the residuals. The ability of a spline to approximate oscillations varies not only with the number of spline sampling points, but also with their exact position. When the distance between two spline sampling points equals exactly or

approximately half the wavelength of the waves present in the data, the last-mentioned effect is most pronounced.

Since knowledge about the wavelengths present in the data set is normally not available in advance, this directly motivates the use of an iterative spline which is based on changing starting points. It comes along with enhanced computational effort but it can be recommended for the approximation / de-trending of individual profiles and if squared residuals are needed for

further analysis (e.g. for the calculation of the wave potential energy).

**Acknowledgement**

We would like to thank the TIMED-SABER team for their great work in providing an excellent data set.
We also thank the Bavarian Ministry for Environment and Consumer Protection for financially supporting our work: V. Wendt was paid by the Bavarian project BHEA (Project number TLK01U-49580, 2010–2013). The work of S. Wüst was subsidised in parts by this project.
At least, we thank Julian Schmoeckel, University of Augsburg, for helping to produce the test data sets and the figures.






## Appendix A

Between two spline sampling points, a spline is equal to a cubic polynomial of the form

$$f(z) = az^3 + bz^2 + cz + d \qquad \text{with} \qquad a, b, c, d \in \mathbb{R}. \tag{2}$$

Its derivatives are

$$f^{(1)}(z) = 3az^2 + 2bz + c, \tag{3}$$

$$f^{(2)}(z) = 6az + 2b, \tag{4}$$

$$f^{(3)}(z) = 6a. \tag{5}$$

Between two spline sampling points, the second derivative of a spline depends linearly on the height coordinate $z$. That means the curvature of the spline can change from negative to positive or vice versa between two spline sampling points but it can only increase or decrease linearly or it can stay constant. At the spline sampling points, all derivatives of the two adjacent polynomials must agree. For example, a spline cannot form two parabolas with different signs in two adjacent intervals in order to approximate a sine / cosine since the second derivative (curvature) would be positive constant in one interval and negative constant in the other. If the spline sampling points are not distributed in a way such that the curvature of the original function increases or decreases linearly between two spline sampling points, the spline cannot approximate the original function properly.

Therefore, the ability of the spline to reproduce a sine / cosine does not only depend on the number of spline sampling points, it varies also with their position.

## Appendix B

The optimal spline parameters are determined through a least-square approach: depending on the spline parameters, the squared differences between the spline and the original data set are minimized. The maximum wavelength which a spline can approximate in principle is equal to two times the distance between two spline sampling points.

Let us denote the oscillation which has to be approximated with $f_1(z)$ and the spline with $f_2(z)$.

If those two oscillations which will be subtracted from each other are characterized by very similar wave numbers $k_1$ and $k_2$, then a beat with the following wave numbers will occur.

$$f_1(z) - f_2(z) = \sin k_1 z - \sin k_2 z \overset{(*)}{=} 2 \cos \left( \frac{k_1 + k_2}{2} z \right) \sin \left( \frac{k_1 - k_2}{2} z \right) \tag{6}$$

where

$\frac{k_1 + k_2}{2}$ is the wave number of the beat, which is very similar to the original wave number, and

$\frac{k_1 - k_2}{2}$ is the wave number of the envelope.

$(*)$: application of an addition theorem




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

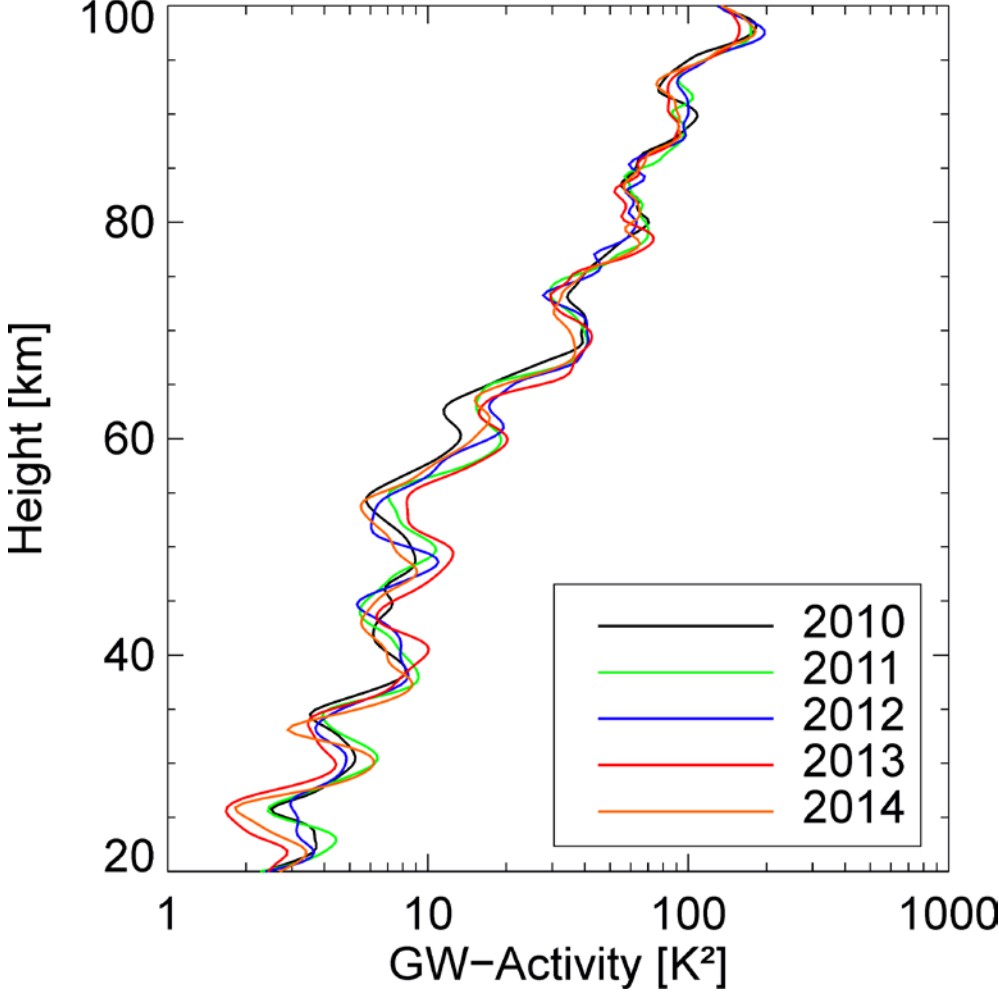

**Figure 1: Mean squared temperature residuals for the years 2010 to 2014 (colour-coded): they are derived from TIMED-SABER, data version 2.0 by using a cubic spline routine with equidistant sampling points for de-trending. The distance between two spline sampling points is 10 km. All vertical SABER temperature profiles which were retrieved between 44° N and 48° N and 5° O and 15° O are used (that means approximately 30-50 profiles per month and approximately 500 profiles per year).**









**Figure 2: This figure shows the approximation of a cubic spline using different numbers of spline sampling points:**

**(a) A spline with a distance of 1.5 km between two spline sampling points is fitted (solid line) to the test data (dotted line).**

**(b) Same as (a) but the distance between two spline sampling points is 1.6 km.**

**(c) Same as (a) but the distance between two spline sampling points is 1.4 km.**

5  **(d) Same as (a) but restricted to the height range between 15 km and 50 km. The dashed-dotted lines refer to the height-coordinate of the spline sampling points. The asterisks show the sampling points of to the original sine.**

**(e) Same as (b) but restricted to the height range between 15 km and 50 km.**

**(f) Same as (c) but restricted to the height range between 15 km and 50 km.**



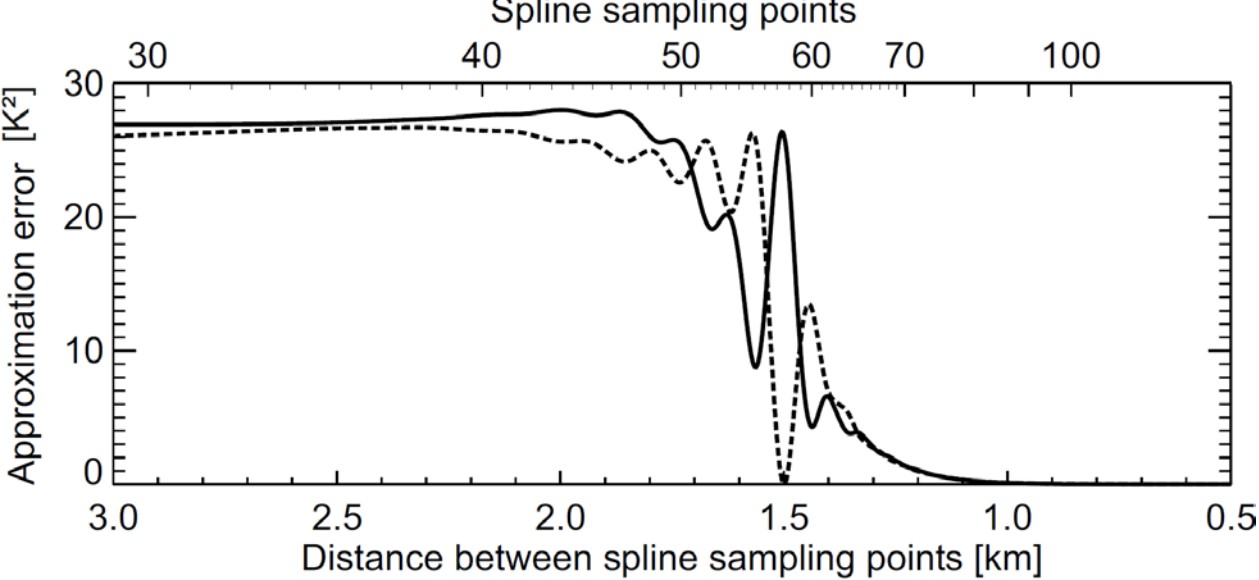

Figure 3: This figure shows the differences between the spline and the approximated test data (solid line: phase of 0, dashed line: phase of π/2) which are summed up between 20 km and 40 km. They are plotted versus the distance between the number of spline sampling points / distance between spline sampling points. The number of spline sampling points / distance between spline sampling points refers to the whole height range between 15 km and 100 km.





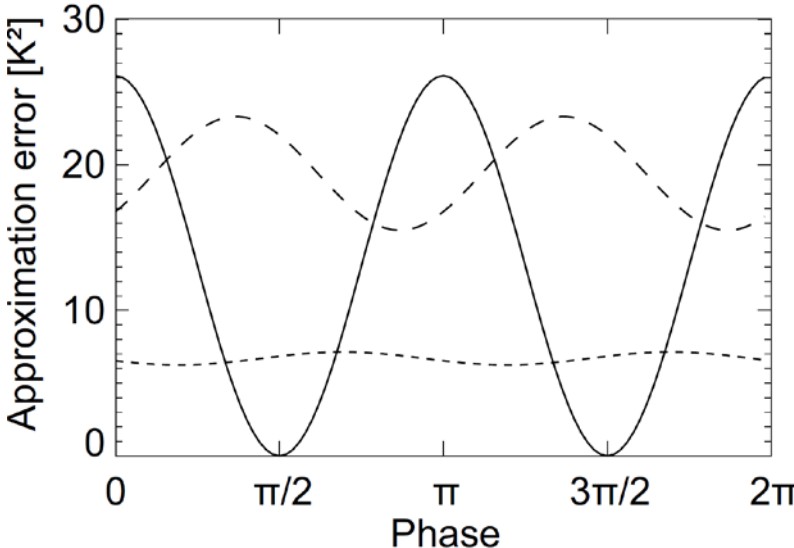

**Figure 4: Dependence of the approximation error on the phase of the wave with a wavelength of 3.0 km and a distance of 1.4 km (short dashes), 1.5 km (solid line) and 1.6 km (long dashes) between two spline sampling points.**







**Figure 5:** Here, the results based on the iterative spline approach are shown. The different colours refer to the different spline approximations (to keep it as clear as possible, we only show the first four iterations, a fifth one exists for case (b) and (e), see step 4 of the algorithm). The black line represents the final spline approximation. The distance between two spline sampling points in part (a) to (f) agrees with the respective values in figure 2 part (a) to (f). While part (a) to (c) show the height range between 15 km and 100 km, part (d) to (f) focus on the height range between 15 km to 50 km. The asterisks and dashed-dotted lines have the same meaning as in figure 2





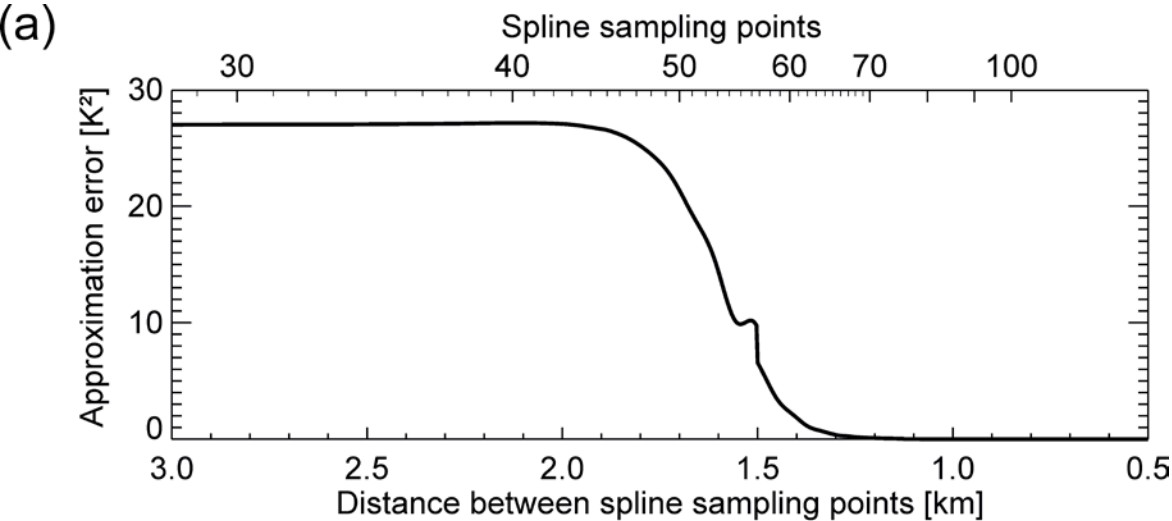

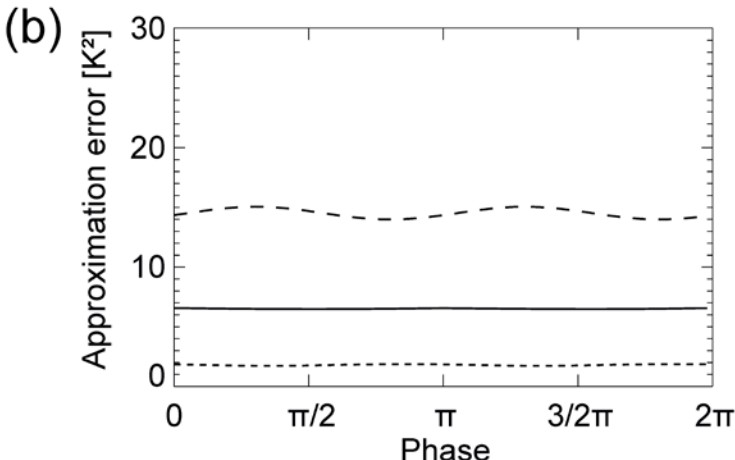

5   **Figure 6: Part (a) is equivalent to figure 3, part (b) is equivalent to figure 4, but here the iterative spline approach is used.**





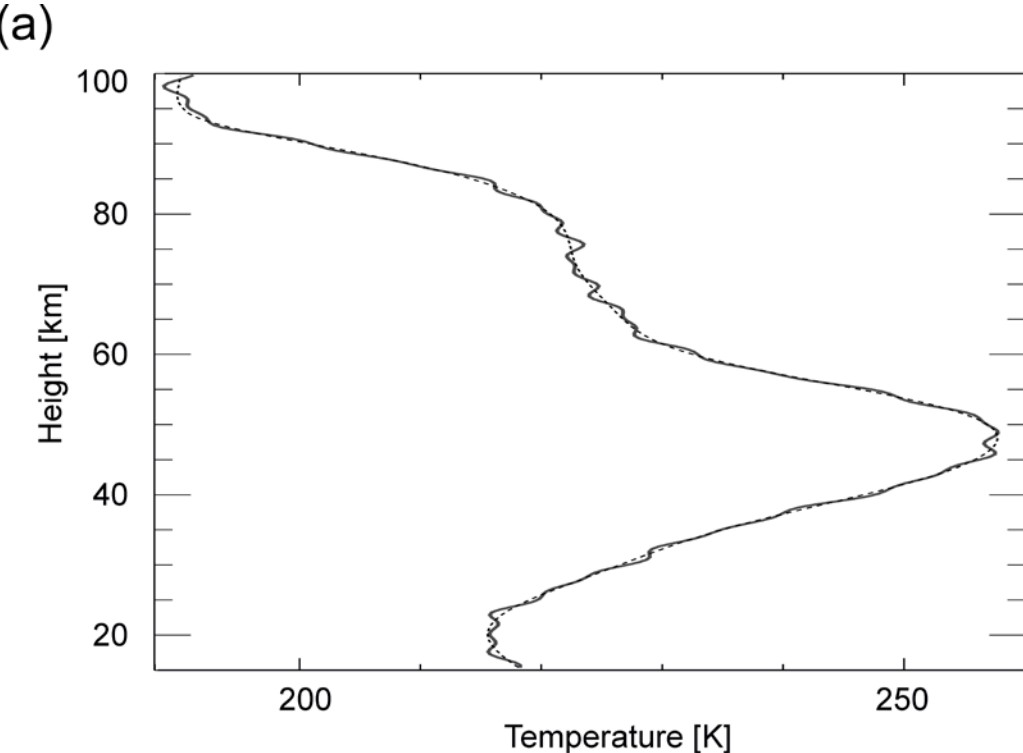

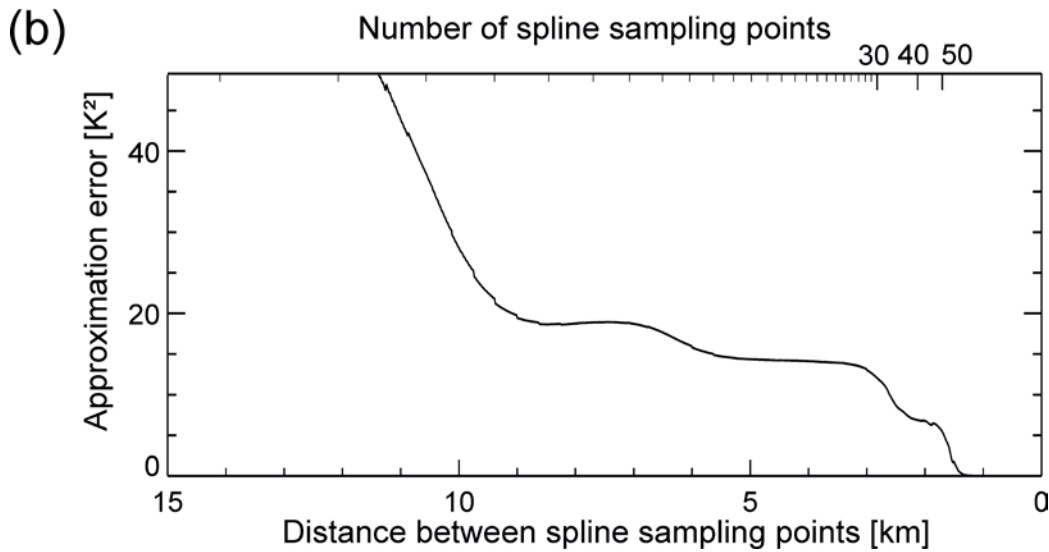

**Figure 7: a) The solid line depicts three sinusoidals with vertical wavelengths of 3, 5 and 13 km, phase 0, π/3, and π /5, and amplitude 0.5 which are superimposed on a realistic temperature profile (dashed line). Part b) shows the approximation error depending on the distance between two spline sampling points.**





(a)

(b)

(c)

(d)

**Figure 8: a)** As figure 1, for de-trending the iterative cubic spline routine with equidistant sampling points as it is described in section 2 is used. **b)** Mean squared residuals for the iterative (dashed line) and the non-iterative (solid line) approach for the year 2014. **c)** Mean (non-squared) residuals for the *non-iterative* approach for the years 2010 to 2014, and **d)** mean (non-squared) residuals for the *iterative* approach.