# Peer review of "Smoothing data series by means of cubic splines: quality of approximation and introduction of a repeating spline approach"

_Atmospheric Measurement Techniques, 2016_

## Referee Comment (RC1) · Anonymous Referee #2 · 7 Mar 2017

**Review comments for "Smoothing data series by means of cubic splines: quality of approximation and introduction of an iterative spline approach" by Sabine Wüst et al.**

The current paper introduces an alternative way of applying cubic splines for smoothing data series. Based on simple test data, the authors first showed the dependence of the approximation error on the sampling distance for the traditional way of applying cubic splines (Fig. 3). It was shown that when the sampling distance is equal or close to half of the wavelength, the approximation error does not decrease smoothly with decrease of the sampling distance, but oscillates strongly. In this case, the approximation error also depends very strongly on the phase of the wave/ exact positions of the sampling points (Fig. 4). In addition, artifical oscillations can appear in this case (Fig. 2b, c).

The authors suggested an alternative method (iterative approach) of applying cubic splines, which can reduce these phenomena. Indeed, using this iterative approach, the approximation error decreases quite smoothly with decrease of the sampling distance (Fig. 6a). Moreover, the dependence of the approximation error on exact positions of sampling points also reduces significantly (Fig. 6b). However, it should be noted that this improvement is only helpful for the sampling distances, which are equal or close to half of the wavelength. Another important note is that the amplitude of the smoothed series (and therefore of the residuals) changes significantly.

The iterative approach was also applied to another simple test dataset, which was created by superimposing 3 oscillations with different wavelengths on a realistic temperature background profile. Again, the approximation error decreases smoothly with decrease of the sampling distance. Finally, the iterative approach was applied to vertical temperature profiles measured by the SABER instrument. The residuals are then calculated and averaged over one year for the years 2010-2014. Results and their implication for gravity wave (GW) studying was discussed.

The paper is generally well written and has a clear structure. Introduction, Method description and Discussion are of appropriate length. My major and minor comments for this paper are given below:

**Major comments**

1. The iteractive approach has advantages of stabilizing the traditional approach: (1) the approximation error decreases smoothly with decrease of the sampling distance, and (2) the approximation error depends very slightly on the exact locations of sampling points. However, these advantages are only helpful if the sampling distance is equal or close to half of the wavelength.

   In practice, a constant sampling distance is chosen for all measurement profiles (for example, the authors chose 10 km for all SABER profiles in this study). It means the iteractive approach may be advanced for only some profiles, which contain waves with wavelength of ∼ 20 km. For other waves in the same profile or for other waves in other profiles, which have wavelengths different from ∼ 20 km, I would not expect advantages of the iteractive approach. It is well known that GWs have a very broad spectrum. Therefore, wavelengths, which are equal or close to a certain wavelength, can only be a very small part of that broad spectrum.

   It is therefore interesting to see how much this stabilizing contributes to the total approximation error:

(a) Starting with the test data, where the 3 oscillations of different wavelengths were superimposed on a realistic CIRA temperature background, can you please compare the approximation error of the traditional approach and the approximation error of the iterative approach? (Using a constant sampling distance of 10 km, for example)

(b) Similarly for SABER profiles per year: For each profile, please calculate the approximation error $\delta_{1i}$ for traditional approach and $\delta_{2i}$ for iterative approach, using a constant sampling distance of 10 km. For one year, the total approximation error is $\Delta_1 = \sum_i \delta_{1i}$ for traditional approach and $\Delta_2 = \sum_i \delta_{2i}$ for iterative approach ($\sum_i$ means sum over all profiles of one year). Can you please compare $\Delta_1$ and $\Delta_2$ for the years 2010-2014?

2. The term called "approximation error" by the authors has the meaning of error in the case study, where the test data are described by an artificial sine. If an oscillation is superimposed on a realistic CIRA temperature background, the error of GW activity should be the difference between the residual (after the spline fit) and the original oscillation you used for superimposing. How does this error of GW activity vary with sampling distance for non-iterative and iterative approaches, for different superimposed wavelengths?

3. When we average the temperature residuals over the whole year for enough number of profiles, due to the arbitrary distribution of phases and amplitudes of GWs, the non-squared mean residual should be approximately zero. However, as shown by Fig. 8d, the iterative method still produces non-zero amplitude oscillations in the temperature residuals. In page 8, lines 20-22, the authors suggested that changing in temperature gradients could be one of the reasons for this problem. However, if it is the main reason, we should see non-zero amplitude oscillations only in limited altitude regions near the stratopause or mesopause, but not in the entire altitude range in Fig. 8d. There seem to be systematic errors that have not been removed. Can you please comment on this?

4. The authors recommended the iterative approach for estimating squared residuals for studying GW activity. Perhaps, the most convincing way to demonstrate if the iterative approach is suitable for GW studies, is comparing the GW squared temperature derived by this approach to the one derived by another method, using the same original data. For example, Ern et al. (2011) showed zonal averages of GW squared temperature for SABER measurements. For your method, taking the sampling distance of 10 km allows for vertical wavelengths up to 20 km. This can cover the main part of the GW spectrum in the stratosphere and mesosphere. It would be very interesting to see if the zonal averages provided by this iterative approach is similar to the ones in Ern et al. (2011).

**Minor comments**

1. Page 2, Lines 5-7: "Conclusions about ... sampling points used". The Shannon's sampling theorem described in Appendix 3 of Gubbins, 2004 is about reconstruction of the original time series from its samples. Since you use a constant sampling distance through each entire data series, it is rather the sampling distance between two samples, which decides which shortest wavelength can be resolved. It is rather about the Nyquist theorem than the Shannon's sampling theorem. I would suggest to write straightforward that: "The shortest wavelength/period which can be resolved by the spline is twice of the sampling distance according to the Nyquist theorem."

2. Page 2, Line 15: "between 48°N ...and 15°O". Please shortly explain here why did you choose this region?

3. Page 2, Lines 19-22: "The distance of 10 km ... up to 20 km". At this point, it is not straightforward for all readers to understand why choosing 10 km distance will lead to maximum wavelength of 20 km. Later, in Sect. 2, paragraph 2, you explained this very nicely. I suggest to move the explanation here and shortly refer to it again in Sect. 2 later.

4. Page 2 Line 27 and Page 7, Line 33: For the motivation of this paper, the authors used Fig. 1, where squared temperature residuals were averaged for the whole year. An oscillation with a wavelength of about 10 km is found and the authors suggested that this oscillation is an artefact of the non-iterative spline approach. Howerver, due to seasonal variations of GW sources and of the background wind, GW activity over the same location at a certain altitude can be different for different seasons. Therefore, the oscillation with a wavelength of about 10 km could also be an artefact of averaging. Can you please comment on this?

5. Page 3, Line 12: "Depending on ... approximated". This sentence is not clear to me. I guess the authors mean: "If we want to approximate the entire data series, the length of the data series must be an integer number of the distance between 2 spline sample points. That is why only certain distances between two consecutive spline sampling points can be chosen if the whole data series is approximated". If this is what the authors want to say, I suggest to rewrite this sentence. This also makes the next paragraph become more understandable.

6. Page 5, Line 7: "gravity waves ... ". Please cite the paper of Ern et al. (2011), which provides a comprehensive GW data set derived from SABER measurements.

7. Page 8, Line 20: Please clarify the "height regions"

**Technical corrections**

Page 2, Line 15 and Page 13, Lines 6, 7: °O and 15°O → 5°E and 15°E
Page 7, Line 15: might be surprising → is not straightforward
Page 8, Line 21: one reason → one of the reasons for
Page 19, Line 5: there are no dashed-dotted lines in Fig. 5e, f.

**References**

Ern, M., Preusse, P., Gille, J. C., Hepplewhite, C. L., Mlynczak, M. G., Russell III, J. M., and Riese, M. (2011). Implications for atmospheric dynamics derived from global observations of gravity wave momentum flux in stratosphere and mesosphere. *J. Geophys. Res.*, 116.

---

## Referee Comment (RC2) · Anonymous Referee #1 · 21 Mar 2017

The paper by Wüst et al. deals with the use of cubic splines for smoothing of data series that contain a superposition of (different) gravity waves and a background state. The authors describe the dependence of the quality of the spline fit on the number and location of sampling points in relation to the wavelength and phase of the wave, respectively. In the atmosphere the wavelengths of the observed waves are typically not well known before detailed analysis and not constant across the data set. A distinct phase relation may in this case result in large approximation errors of the spline fit. The authors therefore propose an "iterative" approach with variable phase relation. The new method is applied to temperature data from TIMED/SABER and used to derive gravity wave activities. The manuscript is well written and the figures are instructive. On

the other hand there is some confusion in the structure and content that needs to be resolved before publication. I describe my concerns in detail below.

Major comments

As the authors describe, splines are mainly used for smoothing of a data series. This can be done to extract a "background state" like a temperature profile undisturbed by gravity waves, to produce a smooth data set including only a subset of waves, or to retrieve the residuals that are then treaded as a wave disturbance and examined further. The purpose of spline fitting largely determines the spline parameters and the quantities that should be evaluated. From my point of view the actual manuscript mixes up between a spline for optimal description of a background (and extraction of waves as residuals) and for optimal wave description (minimizing residuals). For example:

- In line 11 (page 1) the authors mention the approximation of undisturbed conditions, but in line 14 they discuss effects of increasing number of sampling points, even if this obviously contradicts the purpose of getting a background state.

- In line 4/5 (page 2) the authors indeed state the goal of getting a "sufficiently low number of sampling points" to describe the background.

- In Section 3 a much shorter sampling point interval compared to the Introduction is used. Accordingly a wave with small vertical wavelength is examined. Even if residuals between the spline fit and the original are calculated, they are minimized ("approximation error") in order to get an optimal reproduction of the wave.

- In the case with CIRA and superposed waves (Section 3 and Figure 7) again a very small sampling point distance is used in the example. If the spline shall be used to extract an "undisturbed" profile, I suggest to calculate the squared difference between the fitted and the original CIRA profile, not between the fitted and the disturbed profile.

- In section 4 (p. 7, l. 2) again the "ability of a spline to approximate oscillations" is described, not the smoothing of the oscillations to retrieve the background.

[Figure]

- In Fig. 8, again a larger sampling point distance is applied and the residuals are examined.

- In the Summary (p. 9, l. 8) the "ability of a spline to approximate oscillations" is mentioned, not the ability to reproduce a background state (and wave-induced residuals).

Please clarify.

Minor comments:

- Introduction: I am surprised that the 10 km oscillations appear so steadily in the averaged data. The later analysis suggests that there is a gravity wave with fixed wavelength and phase that comes out in all altitudes and in all years. This is quite unexpected. Effects of changing temperature gradients (p. 8, l. 20) should appear locally in the profiles.

- P. 3, l. 16-18: This sentence structure is rather complicate. I suggest rephrasing and, e.g., separating in two sentences.

- Section 2 (and others): I find the term "iterative" misleading. Typically it is used for some processes where the adaptive change of parameters produces some converging results. What is done here is that the algorithm averages across all phase differences that are possible between sampling points and the shortest resolvable wave.

- Figure 7 b: It would be interesting to see the error also for the non-interative spline fit.

- P. 7, l. 5: If the wavelength of the sine is so close to the shortest resolvable wavelength, this dependence is not surprising. The problem should be much smaller, if less sampling points are applied and the spline is used to extract the residuals.

- P. 7, l. 25: Is it intended to compare the non-iterative and iterative method (Fig. 3 and 6a) or constant or CIRA background (Fig. 7) as suggested in l. 22. For Fig. 7 it should be noted that the scaling of error and sampling point distance differs from the other plots.

[Figure]

- P. 8, l. 1: I do not see that wave activity is less variable compared to higher altitudes. I only see that amplitudes show a much smaller increase with altitude compare to below and above.

- Section 4: In general it would be helpful to see the results of standard and iterative spline fitting also for a single SABER profile, containing a superposition of background and different waves. This may help to understand the mean profiles of non-squared and squared residuals.

- P. 8, l. 17-18: i) It is still surprising that the annual mean residuals ($\sim$500 profiles) show such a pronounced oscillation. Please comment on this. ii) The amplitudes shown in Fig. 8 c) and d) are non-squared averages and the individual residuals should me much larger. On the other hand the iterative approach should have largest effect if the wavelength is close to double the sampling point distance. This should mainly appear in cases of wave approximation and less in cases of background approximation.

---

## Author Comment (AC2) · 28 Apr 2017

The paper by Wüst et al. deals with the use of cubic splines for smoothing of data series that contain a superposition of (different) gravity waves and a background state. The authors describe the dependence of the quality of the spline fit on the number and location of sampling points in relation to the wavelength and phase of the wave, respectively. In the atmosphere the wavelengths of the observed waves are typically not well known before detailed analysis and not constant across the data set. A distinct phase relation may in this case result in large approximation errors of the spline fit. The authors therefore propose an "iterative" approach with variable phase relation. The new method is applied to temperature data from TIMED/SABER and used to derive gravity wave activities. The manuscript is well written and the figures are instructive. On

the other hand there is some confusion in the structure and content that needs to be resolved before publication. I describe my concerns in detail below.

Major comments

As the authors describe, splines are mainly used for smoothing of a data series. This can be done to extract a "background state" like a temperature profile undisturbed by gravity waves, to produce a smooth data set including only a subset of waves, or to retrieve the residuals that are then treaded as a wave disturbance and examined further. The purpose of spline fitting largely determines the spline parameters and the quantities that should be evaluated. From my point of view the actual manuscript mixes up between a spline for optimal description of a background (and extraction of waves as residuals) and for optimal wave description (minimizing residuals). For example:

- In line 11 (page 1) the authors mention the approximation of undisturbed conditions, but in line 14 they discuss effects of increasing number of sampling points, even if this obviously contradicts the purpose of getting a background state.

- In line 4/5 (page 2) the authors indeed state the goal of getting a "sufficiently low number of sampling points" to describe the background.

- In Section 3 a much shorter sampling point interval compared to the Introduction is used. Accordingly a wave with small vertical wavelength is examined. Even if residuals between the spline fit and the original are calculated, they are minimized ("approximation error") in order to get an optimal reproduction of the wave.

- In the case with CIRA and superposed waves (Section 3 and Figure 7) again a very small sampling point distance is used in the example. If the spline shall be used to extract an "undisturbed" profile, I suggest to calculate the squared difference between the fitted and the original CIRA profile, not between the fitted and the disturbed profile.

- In section 4 (p. 7, l. 2) again the "ability of a spline to approximate oscillations" is described, not the smoothing of the oscillations to retrieve the background.

- In Fig. 8, again a larger sampling point distance is applied and the residuals are examined.

- In the Summary (p. 9, l. 8) the "ability of a spline to approximate oscillations" is mentioned, not the ability to reproduce a background state (and wave-induced residuals).

Please clarify.

Minor comments:

- Introduction: I am surprised that the 10 km oscillations appear so steadily in the averaged data. The later analysis suggests that there is a gravity wave with fixed wavelength and phase that comes out in all altitudes and in all years. This is quite unexpected. Effects of changing temperature gradients (p. 8, l. 20) should appear locally in the profiles.

- P. 3, l. 16-18: This sentence structure is rather complicate. I suggest rephrasing and, e.g., separating in two sentences.

- Section 2 (and others): I find the term "iterative" misleading. Typically it is used for some processes where the adaptive change of parameters produces some converging results. What is done here is that the algorithm averages across all phase differences that are possible between sampling points and the shortest resolvable wave.

- Figure 7 b: It would be interesting to see the error also for the non-interative spline fit.

- P. 7, l. 5: If the wavelength of the sine is so close to the shortest resolvable wavelength, this dependence is not surprising. The problem should be much smaller, if less sampling points are applied and the spline is used to extract the residuals.

- P. 7, l. 25: Is it intended to compare the non-iterative and iterative method (Fig. 3 and 6a) or constant or CIRA background (Fig. 7) as suggested in l. 22. For Fig. 7 it should be noted that the scaling of error and sampling point distance differs from the other plots.

- P. 8, l. 1: I do not see that wave activity is less variable compared to higher altitudes. I only see that amplitudes show a much smaller increase with altitude compare to below and above.

- Section 4: In general it would be helpful to see the results of standard and iterative spline fitting also for a single SABER profile, containing a superposition of background and different waves. This may help to understand the mean profiles of non-squared and squared residuals.

- P. 8, l. 17-18: i) It is still surprising that the annual mean residuals ($\sim$500 profiles) show such a pronounced oscillation. Please comment on this. ii) The amplitudes shown in Fig. 8 c) and d) are non-squared averages and the individual residuals should me much larger. On the other hand the iterative approach should have largest effect if the wavelength is close to double the sampling point distance. This should mainly appear in cases of wave approximation and less in cases of background approximation.

**Answer to**
*In order to facilitate further discussion, we point out numbers of pages of paragraphs in our answer refer to the version with accepted changes as long as not stated otherwise.*
*The attached manuscript with marked changes includes also the changes due to the comments of the other reviewer.*

**Major comments:**

The definition of background or undisturbed background in our sense depends on the addressed research question. It could contain parts of the wave spectra when one focuses one smaller-scale waves for example; in this case one uses only the residuals. Further spectral analysis requires large-scale de-trending in advance (large-scale in relation to the signatures one is interested in), and splines are a common used method for this because they approximate linear as well as wave-like structures. On the other hand, the residuals could contain small-scale non-wave disturbances which disturb the analysis of a specific scientific question and one is only interested in the spline approximation.

We re-formulated the manuscript (see below) to make clear that we are mainly interested in the residuals in common. Nevertheless, one can also learn more about the spline approximation even if the residuals are in the focus.
Additional and in accordance with the comment of the other reviewer, we also avoid the term "approximation error" now in the manuscript which might have contributed to a lack of clearness and use "sum of squared residuals".

1. Former page 1, line 11 & 14: We changed the first paragraph of the introduction to make clear that in general the background state or the residuals—depending on the research question—can be of interest.

2. Former page 2, line 4/5: sentence deleted

3. Section 3: The values for the case studies (without background and with CIRA background) are more or less arbitrarily chosen, however we motivated their choice in the manuscript.
   The effects of the traditional spline approximation are demonstrated in this section for short (3 km) and long (up to 13 km) vertical wavelength (case study without background and with CIRA background) as well as for different sampling point distances which are also comparable with the one chosen in figure 1.

4. Since we do not focus on the optimization of the fitting per se, your comment concerning the CIRA example is obsolete. We just would like to demonstrate that the method is able to filter for a specific part of the wave spectrum. The steps of decrease in the residuals agree very well with the wavelengths.

5. We included "and therefore to filter for a specific part of the wave spectrum" in order to avoid confusion.

6. Since this figure corresponds to figure 1, we choose a comparable sampling point distance.

7. We hope that we have clarified all confusing points mentioned above and leave the formulation in the summary as it was since this meets the purpose of the manuscript.

**Minor comments:**

1. We interpret this 10 km oscillation as an artificial effect of the analysis (p.7 l.28ff and page 9, first paragraph). A persistent gravity wave with fixed wavelength and phase seems also to us very unrealistic.

2. Done.

3. We use the term "iterative" in its original sense (latin: iterare = to repeat), however, we understand your comment that iterative is frequently used today not only for something which is repeated but for something which converges due to adapting certain parameters in the different repetition steps. Since also the other reviewer had problems with this term (he used interactive instead of iterative), it is probably the best choice to avoid this term. We propose to use "repeating spline algorithm" instead of "iterative spline algorithm".

4. Done, annotation: former figure 7b is now 7c.

5. Yes, fig. 3 points out that besides specific conditions (wavelength ≈ sampling difference) there are no problems. However, in practice, atmospheric profiles / data series are often limited in their altitude or time range. So, it is necessary to set the sampling point differences near to the length of present wave structures. Our approach shows that even in these cases it is possible to get useful results. We replaced the term "surprising".

6. Both, the intention of this paragraph is to point out that the repeating approach shows the mentioned features for both backgrounds. We tried to make it clearer in the manuscript. We also added to the figure caption "The range of the x- and y-axis differs from the ones used in fig. 3 and 6 (a)".The scaling of the sum of squared residuals depends on the data resolution since it is the sum over the altitude range 20–40km. The range of the sampling point is extended for the additional wavelengths of 5 and 13 km for the more realistic example in contrast to the pure theoretical examples mentioned before.

7. Improved.

8. Due to the comments of the other reviewer, we included such a comparison already for the CIRA example (in figure 6). Since the main point—the different approximation of traditional and repeating spline—can also be observed in the CIRA example, we hope that it is ok not to show a more or less identical figure here for one SABER profile. Furthermore, we extended the first paragraph of page 9 where we provide possible explanations for the features observed in figure 8.

9. (i) See page 9, first paragraph for possible explanations. The answer to the major comment, number 3 of the other reviewer helps to better understand the information which we added to this paragraph.

(ii) I am not quite sure that I understood every point here correctly but I hope that I can clarify the main points of confusion. Part c and d of figure 8 shows the mean (non-squared) residuals: the mean (non-squared) residuals are calculated as the average over the individual residuals which are indeed much larger and can be positive as well as negative. Due to the latter, we would assume that the mean residual is nearly zero. Obviously, this is not the case: a slight oscillation with an amplitude of ca. 0.5–1 K at 20 km height and ca. 2 K at 100 km height can be observed (in figure 8c and d, logarithmic scaling) for which physical reasons do not exist from our point of view and which is presumably an artefact of the analysis (see page 9, first paragraph for possible explanations).

[revised manuscript text omitted]

---

## Author Response (AR1)

**Answer to**

**Review comments for "Smoothing data series by means of cubic splines: quality of approximation and introduction of an iterative spline approach" by Sabine Wüst et al.**

*First of all, we would like to thank you for your valuable comments.*
*In order to avoid confusion, we would like to mention already here that we proposed to use "repeating" spline algorithm since the other reviewer had problems with the term "iterative".*
*The attached manuscript with marked changes includes also the changes due to the comments of the other reviewer.*

**Major comments:**

**1.**
**(a)** Starting with the test data, where the 3 oscillations of different wavelengths were superimposed on a realistic CIRA temperature background, can you please compare the approximation error of the traditional approach and the approximation error of the iteractive approach? (Using a constant sampling distance of 10 km, for example)

As proposed, we compare the approximation error of both approaches based on the test data, where the three oscillations of different wavelengths were superimposed on a realistic CIRA temperature background. In all subfigures of figure 7, additional curves for the traditional approach are added. In order to make the pictures easier to read, we changed the wave amplitudes from 0.5 to 2.

**(b)** Similarly for SABER profiles per year: For each profile, please calculate the approximation error $\delta_{1i}$ for traditional approach and $\delta_{2i}$ for iteractive approach, using a constant sampling distance of 10 km. For one year, the total approximation error is $\Delta_1 = \sum_i \delta_{1i}$ for traditional approach and $\Delta_2 = \sum_i \delta_{2i}$ for iteractive approach ($\sum_i$ means sum over all profiles of one year). Can you please compare $\Delta_1$ and $\Delta_2$ for the years 2010-2014?

We already provided the sum of squared residuals which we denote as approximation error in figure 8) and b) as well as in figure 1 for a constant sampling point distance of 10 km.

**2.** The term called "approximation error" by the authors has the meaning of error in the case study, where the test data are described by an artificial sine. If an oscillation is superimposed on a realistic CIRA temperature background, the error of GW activity should be the difference between the residual (after the spline fit) and the original oscillation you used for superimposing. How does this error of GW activity vary with sampling distance for non-iteractive and iteractive approaches, for different superimposed wavelengths?

Actually, the dashed line in figure 7a describes a spline approximation (repeating approach, difference between spline sampling points 10 km) of the background and not the original background. We are sorry for this non-optimal figure description in the original manuscript. However, one can see that the background is already very well adapted for this sampling point distance. Main reason for the variation of the "approximation error" is the increasing adaption of the

superimposed oscillations (see also figure 5 and 6). When calculating the difference between the residuals and the original oscillation for different sampling point distances, the corresponding does not look not very instructive any more from our point of view. When the spline is able to approximate one of the original oscillations, the residuals do not contain this oscillation any more. Subtracting the original oscillation from the residuals in this case, leads to a larger difference than before. That means "your" approximation error grows again.

Since the term "approximation error" is ambiguous (also the other reviewer seems having problems with it, see point 1 of his major comments), we leave it out now in the entire manuscript and call it sum of squared residuals.

**3.** When we average the temperature residuals over the whole year for enough number of profiles, due to the arbitrary distribution of phases and amplitudes of GWs, the non-squared mean residual should be approximately zero. However, as shown by Fig. 8d, the iterative method still produces non-zero amplitude oscillations in the temperature residuals. In page 8, lines 20-22, the authors suggested that changing in temperature gradients could be one of the reasons for this problem. However, if it is the main reason, we should see non-zero amplitude oscillations only in limited altitude regions near the stratopause or mesopause, but not in the entire altitude range in Fig. 8d. There seem to be systematic errors that have not been removed. Can you please comment on this?

We made some further tests in order to answer your question. We de-trended the CIRA temperature values between ca. 14 and 110 km height (as we did it for the SABER data) with three different sampling point distances (2.5 km, 5 km and 10 km). CIRA data do not show small-scale fluctuations and so we can test how well the spline adaption works for a smooth background. For the sampling point distances of 2.5 and 5 km, it becomes clear that the amplitude of the oscillation shows local maxima at the upper and lower border (see figure 1). For a sampling pont distance of 10 km, the oscillation covers the whole height range. We add this information to the manuscript (page 10 first paragraph).

[Figure]

**Figure 1** *First panel: CIRA-temperature profile which was detrended between 14 and 110 km height (as the SABER data). Second to fourth panel: Residuals for a sampling point distance of 2.5 k, 5 km and 10 km between 20 and 100 km height. Please be aware of the changing x-scale.*

**4.** The authors recommended the iterative approach for estimating squared residuals for studying GW activity. Perhaps, the most convincing way to demonstrate if the iterative approach is suitable for GW studies, is comparing the GW squared temperature derived by this approach to the one derived by another method, using the same original data.

For example, Ern et al. (2011) showed zonal averages of GW squared temperature for SABER measurements. For your method, taking the sampling distance of 10 km allows for vertical wavelengths up to 20 km. This can cover the main part of the GW spectrum in the stratosphere and mesosphere. It would be very interesting to see if the zonal averages provided by this iteractive approach is similar to the ones in Ern et al. (2011).

We agree that the most convincing argument for this approach would be a comparison to other ones. We choose Shuai et al. (2014). Different to Ern et al. (2011), which you proposed, those authors provide the same parameter as we do—squared temperature fluctuations. Ern et al. (2011) use gravity wave amplitudes which stand in relation to the squared temperature fluctuations but this makes the comparison unnecessarily more complicated.
Shuai et al. (2014) use an earlier version of TIMED-SABER data (1.07) and a different de-trending procedure as we do. In their figure 2, they show monthly averages of the squared temperature fluctuations for the years 2002–2010. Different to us, they provide this parameter in dB ($10 \cdot log_{10} (T'^2_{GW})$) with the squared temperature fluctuation $T'^2_{GW}$ which makes a short calculation necessary.

For 100 km height, we extract a yearly mean of ca. 21 dB for 50°N from their figure 2 that means
$$\cdot log_{10} (T'^2_{GW}) = 21 \Leftrightarrow log_{10} (T'^2_{GW}) = 2.1 \Leftrightarrow T'^2_{GW} = 10^{2.1} \approx 126$$

For 25 km height, we read am mean value of ca. 4 dB:
$$\cdot log_{10} (T'^2_{GW}) = 4 \Leftrightarrow log_{10} (T'^2_{GW}) = 0.4 \Leftrightarrow T'^2_{GW} = 10^{0.4} \approx 2.5$$

These values agree very well with the ones provided by us, however the colour-code used in Shuai et al. (2014) makes it difficult to give a profound answer to the question which spline approach agrees better. Nevertheless, we can argue that the overall structure which is characterized by a nearly constant or slow increase of gravity wave activity in the upper stratosphere can be observed in their figure 2 and also our figure 8a. We include this comparison also in the manuscript.

**Minor comments:**

**1.** Page 2, Lines 5-7: "Conclusions about ... sampling points used". The Shannon's sampling theorem described in Appendix 3 of Gubbins, 2004 is about reconstruction of the original time series from its samples. Since you use a constant sampling distance through each entire data series, it is rather the sampling distance between two samples, which decides which shortest wavelength can be resolved. It is rather about the Nyquist theorem than the Shannon's sampling theorem. I would suggest to write straightforward that: "The shortest wavelength/period which can be resolved by the spline is twice of the sampling distance according to the Nyquist theorem."

Corrected in the entire manuscript.

**2.** Page 2, Line 15: "between 48_N ...and 15_O". Please shortly explain here why did you choose this region?

Done.

**3.** Page 2, Lines 19-22: "The distance of 10 km ... up to 20 km". At this point, it is not straightforward for all readers to understand why choosing 10 km distance will lead to maximum wavelength of 20 km. Later, in Sect. 2, paragraph 2, you explained this very nicely. I suggest to move the explanation here and shortly refer to it again in Sect. 2 later.

Done.

**4.** Page 2 Line 27 and Page 7, Line 33: For the motivation of this paper, the authors used Fig. 1, where squared temperature residuals were averaged for the whole year. An oscillation with a wavelength of about 10 km is found and the authors suggested that this oscillation is an artefact of the non-iterative spline approach. However, due to seasonal variations of GW sources and of the background wind, GW activity over the same location at a certain altitude can be different for different seasons. Therefore, the oscillation with a wavelength of about 10 km could also be an artefact of averaging. Can you please comment on this?

[Figure]

*Figure 2 Both, the yearly mean of the year 2014 and also the mean over January 2014, show oscillations (repeating spline approach).*

You are right, after figure 1 we cannot exclude that the oscillation is an artefact of averaging over the whole year. However, we made exactly the same analysis with the repeating approach. We did not change the data basis or anything else, we just used the other spline approach and the oscillation became less pronounced. This contradicts the assumption that the oscillation is an

artefact of averaging. Additionally, we had a look into monthly data but the result is the same (see figure 2)

**5.** Page 3, Line 12: "Depending on ... approximated". This sentence is not clear to me. I guess the authors mean: "If we want to approximate the entire data series, the length of the data series must be an integer number of the distance between 2 spline sample points. That is why only certain distances between two consecutive spline sampling points can be chosen if the whole data series is approximated". If this is what the authors want to say, I suggest to rewrite this sentence. This also makes the next paragraph become more understandable.

This is exactly what we meant. We re-formulated this paragraph.

**6.** Page 5, Line 7: \gravity waves ... ". Please cite the paper of Ern et al. (2011), which provides a comprehensive GW data set derived from SABER measurements.

Done.

**7.** Page 8, Line 20: Please clarify the "height regions"

We mean the tropo-, strato-, and mesopause corrected in the text.

**Technical corrections:**

**1.** Page 2, Line 15 and Page 13, Lines 6, 7: 5°O and 15°O → 5°E and 15°E.
Corrected.

**2.** Page 7, Line 15: might be surprising → is not straightforward
You probably meant line 5, which we re-formulated.

**3.** Page 8, Line 21: one reason → one of the reasons for
Corrected.

**4.** Page 19, Line 5: there are no dashed-dotted lines in Fig. 5e, f.
Corrected

**Answer to**
*In order to facilitate further discussion, we point out numbers of pages of paragraphs in our answer refer to the version with changes as long as not stated otherwise.*
*The attached manuscript with marked changes includes also the changes due to the comments of the other reviewer.*

**Major comments:**

As the authors describe, splines are mainly used for smoothing of a data series. This can be done to extract a "background state" like a temperature profile undisturbed by gravity waves, to produce a smooth data set including only a subset of waves, or to retrieve the residuals that are then treaded as a wave disturbance and examined further. The purpose of spline fitting largely determines the spline parameters and the quantities that should be evaluated. From my point of view the actual manuscript mixes up between a spline for optimal description of a background (and extraction of waves as residuals) and for optimal wave description (minimizing residuals).

The definition of background or undisturbed background in our sense depends on the addressed research question. It could contain parts of the wave spectra when one focuses one smaller-scale waves for example; in this case one uses only the residuals. Further spectral analysis requires large-scale de-trending in advance (large-scale in relation to the signatures one is interested in), and splines are a common used method for this because they approximate linear as well as wave-like structures. On the other hand, the residuals could contain small-scale non-wave disturbances which disturb the analysis of a specific scientific question and one is only interested in the spline approximation.

We re-formulated the manuscript (see below) to make clear that we are mainly interested in the residuals in common. Nevertheless, one can also learn more about the spline approximation even if the residuals are in the focus.
Additional and in accordance with the comment of the other reviewer, we also avoid the term "approximation error" now in the manuscript which might have contributed to a lack of clearness and use "sum of squared residuals".

**1.** In line 11 (page 1) the authors mention the approximation of undisturbed conditions, but in line 14 they discuss effects of increasing number of sampling points, even if this obviously contradicts the purpose of getting a background state.

We changed the first paragraph of the introduction to make clear that in general the background state or the residuals—depending on the research question—can be of interest.

**2.** In line 4/5 (page 2) the authors indeed state the goal of getting a "sufficiently low number of sampling points" to describe the background.

Sentence deleted

**3.** In Section 3 a much shorter sampling point interval compared to the Introduction is used. Accordingly a wave with small vertical wavelength is examined. Even if residuals between the spline fit and the original are calculated, they are minimized ("approximation error") in order to get an optimal reproduction of the wave.

The values for the case studies (without background and with CIRA background) are more or less arbitrarily chosen, however we motivated their choice in the manuscript.
The effects of the traditional spline approximation are demonstrated in this section for short (3 km) and long (up to 13 km) vertical wavelength (case study without background and with CIRA background) as well as for different sampling point distances which are also comparable with the one chosen in figure 1.

**4.** In the case with CIRA and superposed waves (Section 3 and Figure 7) again a very small sampling point distance is used in the example. If the spline shall be used to extract an "undisturbed" profile, I suggest to calculate the squared difference between the fitted and the original CIRA profile, not between the fitted and the disturbed profile.

Since we do not focus on the optimization of the fitting per se, your comment concerning the CIRA example is obsolete. We just would like to demonstrate that the method is able to filter for a specific part of the wave spectrum. The steps of decrease in the residuals agree very well with the wavelengths.

**5.** In section 4 (p. 7, l. 2) again the "ability of a spline to approximate oscillations" is described, not the smoothing of the oscillations to retrieve the background.

We included "and therefore to filter for a specific part of the wave spectrum" in order to avoid confusion.

**6.** In Fig. 8, again a larger sampling point distance is applied and the residuals are examined.

Since this figure corresponds to figure 1, we choose a comparable sampling point distance.

**7.** In the Summary (p. 9, l. 8) the "ability of a spline to approximate oscillations" is mentioned, not the ability to reproduce a background state (and wave-induced residuals).

We hope that we have clarified all confusing points mentioned above and leave the formulation in the summary as it was since this meets the purpose of the manuscript.

**Minor comments:**

**1.** Introduction: I am surprised that the 10 km oscillations appear so steadily in the averaged data. The later analysis suggests that there is a gravity wave with fixed wavelength and phase that comes out in all altitudes and in all years. This is quite unexpected. Effects of changing temperature gradients (p. 8, l. 20) should appear locally in the profiles.

We interpret this 10 km oscillation as an artificial effect of the analysis (see e.g. p. 8 last paragraph). A persistent gravity wave with fixed wavelength and phase seems also to us very unrealistic.

**2.** P. 3, l. 16-18: This sentence structure is rather complicate. I suggest rephrasing and, e.g., separating in two sentences.

Done.

**3.** Section 2 (and others): I find the term "iterative" misleading. Typically it is used for some processes where the adaptive change of parameters produces some converging results. What is done here is that the algorithm averages across all phase differences that are possible between sampling points and the shortest resolvable wave.

We use the term "iterative" in its original sense (latin: iterare = to repeat), however, we understand your comment that iterative is frequently used today not only for something which is repeated but for something which converges due to adapting certain parameters in the different repetition steps. Since also the other reviewer had problems with this term (he used interactive instead of iterative), it is probably the best choice to avoid this term. We propose to use "repeating spline algorithm" instead of "iterative spline algorithm".

**4.** Figure 7 b: It would be interesting to see the error also for the non-interative spline fit.

Done, annotation: former figure 7b is now 7c.

**5.** P. 7, l. 5: If the wavelength of the sine is so close to the shortest resolvable wavelength, this dependence is not surprising. The problem should be much smaller, if less sampling points are applied and the spline is used to extract the residuals.

Yes, fig. 3 points out that besides specific conditions (wavelength ≈ sampling difference) there are no problems. However, in practice, atmospheric profiles / data series are often limited in their altitude or time range. So, it is necessary to set the sampling point differences near to the length of present wave structures. Our approach shows that even in these cases it is possible to get useful results. We replaced the term "surprising".

**6.** P. 7, l. 25: Is it intended to compare the non-iterative and iterative method (Fig. 3 and 6a) or constant or CIRA background (Fig. 7) as suggested in l. 22.

For Fig. 7 it should be noted that the scaling of error and sampling point distance differs from the other plots.

Both, the intention of this paragraph is to point out that the repeating approach shows the mentioned features for both backgrounds. We tried to make it clearer in the manuscript. We also added to the figure caption "The range of the x- and y-axis differs from the ones used in fig. 3 and 6 (a)".The scaling of the sum of squared residuals depends on the data resolution since it is the sum over the altitude range 20–40km. The range of the sampling point is extended for the additional wavelengths of 5 and 13 km for the more realistic example in contrast to the pure theoretical examples mentioned before.

**7.** P. 8, l. 1: I do not see that wave activity is less variable compared to higher altitudes. I only see that amplitudes show a much smaller increase with altitude compare to below and above.

Improved.

**8.** Section 4: In general it would be helpful to see the results of standard and iterative spline fitting also for a single SABER profile, containing a superposition of background and different waves. This may help to understand the mean profiles of non-squared and squared residuals.

Due to the comments of the other reviewer, we included such a comparison already for the CIRA example (in figure 6). Since the main point—the different approximation of traditional and repeating spline—can also be observed in the CIRA example, we hope that it is ok not to show a more or less identical figure here for one SABER profile. Furthermore, we extended the first paragraph of page 10 where we provide possible explanations for the features observed in figure 8.

**9. (i)** P. 8, l. 17-18: i) It is still surprising that the annual mean residuals (~500 profiles) show such a pronounced oscillation. Please comment on this.

See page 10, first paragraph for possible explanations. The answer to the major comment, number 3 of the other reviewer helps to better understand the information which we added to this paragraph.

**ii)** The amplitudes shown in Fig. 8 c) and d) are non-squared averages and the individual residuals should me much larger. On the other hand the iterative approach should have largest effect if the wavelength is close to double the sampling point distance. This should mainly appear in cases of wave approximation and less in cases of background approximation.

I am not quite sure that I understood every point here correctly but I hope that I can clarify the main points of confusion. Part c and d of figure 8 shows the mean (non-squared) residuals: the mean (non-squared) residuals are calculated as the average over the individual residuals which are indeed much larger and can be positive as well as negative. Due to the latter, we would assume that the mean residual is nearly zero. Obviously, this is not the case: a slight oscillation with an amplitude of ca. 0.5– 1 K at 20 km height and ca. 2 K at 100 km height can be observed (in figure 8c and d, logarithmic scaling) for which physical reasons do not exist from our point of view and which is presumably an artefact of the analysis (see page 10, first paragraph for possible explanations).

[revised manuscript text omitted]

---

## Author Response (AR2)

**Answer to Report #1**

Thank you very much for your valuable comments. I answered them in the following and changed the manuscript accordingly especially at the end of section 4 where the reasons for the oscillation in the mean squared residuals are discussed.

The revised version of the Wüst et al. manuscript is strongly improved compared to the original version. The authors clarified the scope of their method and adapted the description according to both reviewers' comments. I am happy that the scope is much clearer now. Using the spline method for the description of the background and taking the residuals as wave disturbance is a widely used approach in data analyses. Therefore, it has been useful, e.g., to change the term "approximation error" into "sum of squared residuals". Nevertheless I am still concerned about the structure of the manuscript. Furthermore, a few answers to my previous review stimulated some new comments.

**General comments:**

1.  The authors made clear that the description of (gravity) waves from the residuals of the spline fit is the major topic of this paper. In detail, the presumably artificial oscillation in averaged SABER GW activity is the main motivation to develop improved methods of spline fitting. On the other hand they only shortly speculate about the true origin of this oscillation. Instead, they state in the Discussion that the analysis "is beyond the scope of this manuscript". I suggest to at least try to examine the true reason for this oscillation. Is it visible in every profile or in monthly averages? Is it only a result of averaging? Does it also appear if some random GW are added to a CIRA profile? Does it also appear if the sampling point distance is set to, e.g., 15 km? See also below my addition to the old comment #9.
    I included two more figures (9 and 10, one of them figure 1 of the former answer to the reviewer comments) and discussed them at the end of section 4 in order to make clearer where the mentioned oscillations probably come from.

2.  From my point of view it should be made clearer that the difference between normal and repeating spline fitting is largest if the sampling point distance is close to the half wavelength.
    I added this information in the last paragraph of section 4.
    In case of the constant background the differences between both methods vanish for distances larger than 2 km or smaller than about 1.3 km. In case of the CIRA background, differences are always only about 20% of the sum of squared residuals. Here it would be very interesting to see, which of both methods compares better with the true background profile. This comparison can only be done with CIRA because here the background is exactly known.
    I included this comparison in figure 7 (d)

3.  From my point of view, the essential information from this paper is twofold:
    i) The spline method may produce some artefacts in separating background and waves if the data set contains waves with wavelengths of about double the sampling point distance. This can at least partly be overcome with the repeating spline.
    I pointed this out in the last paragraph of section 4 and the summary.
    ii) The description of the background by a too coarse spline may produce artificial oscillations in the residuals which should not be confused with GW. This interesting result from the answer to Reviewer 1 should also be presented in the paper, because it demonstrates the limitations of (non-repeating) splines in general, if the background is unknown (as usual).
    I added this information in the summary.
    Further concerns about the structure of the manuscript are described in the specific comments below.

**Specific Comments to the Author Response:**

1. Old Comment 7: In the Summary (p. 9, l. 8) the "ability of a spline to approximate oscillations" is mentioned, not the ability to reproduce a background state (and wave-induced residuals).
   Answer: We hope that we have clarified all confusing points mentioned above and leave the formulation in the summary as it was since this meets the purpose of the manuscript.
   My comment: At different positions of the manuscript you write that the motivation is to remove the oscillations in the wave activity (Fig. 1). Obviously this can only be done by improved description of the background by the spline, not by optimal approximation of the oscillations.
   Changed, see answer to specific comment no. 8

2. Old Comment 9. (i) P. 8, l. 17-18: i) It is still surprising that the annual mean residuals (~500 profiles) show such a pronounced oscillation. Please comment on this.
   Answer: See page 10, first paragraph for possible explanations. The answer to the major comment, number 3 of the other reviewer helps to better understand the information which we added to this paragraph.
   My comment: As written above, the new figure provided in the answer to Reviewer 1 would also improve the manuscript. It shows limitations of the standard spline approximation by an impressive example. Furthermore, it gives some hints on the question why oscillations with a wavelength similar to the sampling point distance appear in the average data of Fig. 1, but may not be visible in a single GW profile.
   See answer to general comment no. 1

**Specific Comments to the manuscript (line numbers refer to the manuscript without tracked changes):**

1. p.1, l. 14: Maybe I am too strict on this, but I suggest writing "approximation of the background state"
   Done

2. Section 3, case study with constant background: Obviously in this section it is intended to show a method for a good approximation of the oscillations (disturbed profile), see, e.g., p. 5, l. 27, "sinusoidal approximation is better adapted". This is somewhat outside of the main motivation. Therefore it should be made clear that, e.g., this section helps to understand the general behavior of splines if the data set contains waves with a wavelength of double the sampling point distance – which may happen in the general case of an unknown mixture of waves.
   We provide the information about the purpose of this section now right at the section beginning. Furthermore, we deleted the sentence "sinusoidal approximation is better adapted" and re-formulated the sentence "the oscillation is adapted very well between 20 km and 40 km for a phase of $\pi/2$ and $3\pi/2$." (p. 6, ll. 2–3, version with changes marked)

3. Section 3, case study with CIRA: Now it seems that a good approximation of the original CIRA profile is intended. Therefore, it would be very helpful to show also the original profile in Fig. 7. This helps the reader to estimate whether the conventional spline or the repeating spline removes the waves better. Additionally, also the residuals between both retrieved profiles and the original CIRA should be calculated.
   I included the original CIRA-profile in figure 7 a & b. I also plotted the difference between the both retrieved profiles and the original CIRA-background (see figure 7d) and changed the manuscript accordingly (p6. ll.19–22, version with changes marked).

4. p. 7, l. 1: Again, I thought that the motivation is not to reproduce oscillations by the spline, but to reproduce the background and to calculate the GW from the residuals.

I changed the sentence into "In section 3, we showed that the quality of spline to approximate the background and also its ability to filter for a specific part of the wave spectrum varies" to point out that the purpose is the background approximation including from case to case also large-scale gravity wave signature for example.

5. p. 8, l. 8-13: The calculation is straightforward. I suggest shortening this part and giving the numbers in dB and K^2 without further explanation.
Done

6. p. 8, l. 14-16: I think it is hard to say from the comparison with Shuai et al., which method is better. Their numbers are comparable to both, Fig. 1 and Fig. 8, especially if potential longitudinal variations are taken into account. I suggest adding an appropriate statement.
I changed "These values agree very well with the ones provided here" to "These values agree very well with the ones provided here but it cannot be decided whether they match better with the ones based on the repeating or non-repeating approach (fig. 1 or 8 (a))". Unfortunately I do not have access to the Shuai et al. paper. If they show profiles, it would be helpful to learn whether these are also showing the "surprising" oscillations as Fig. 1. Shuai et al. provide exclusively colour-coded means (figure 2) or profiles of the maximum of monthly averaged GW activity over all years (figure 3). The latter shows fluctuations but they can also be due to the fact that the look for the maximum at each height. Due to the scaling of the colour bar and the use of dB (which includes a logarithm and makes fluctuations appearing smaller) it is from my point not possible to provide a serious statement about possible fluctuations.

7. p. 9, l. 3: From my understanding, the motivation of this paper is to remove these oscillations (see Introduction and other places). Therefore, the analysis should not be "beyond the scope". See above.
See answer above.

8. p. 10, l. 8: I suggest (again) rephrasing "ability of the spline to approximate oscillations". See above.
I changed "oscillations" to the "background state (and large-scale wave-induced fluctuations)".

**Typos**:
1. p. 9, l. 1: "not very much" should read "not vary much"
Done

2. Fig. 6: "reisduals" should read "residuals"
Done

[revised manuscript text omitted]